# The Interconnection Between Systemic Lupus Erythematosus and Diet: Unmet Needs, Available Evidence, and Guidance—A Patient-Driven, Multistep-Approach Study

**DOI:** 10.3390/nu16234132

**Published:** 2024-11-29

**Authors:** Savino Sciascia, Gabriele Ferrara, Lorenzo Roccatello, Elena Rubini, Silvia Grazietta Foddai, Massimo Radin, Irene Cecchi, Daniela Rossi, Alice Barinotti, Fulvio Ricceri, Winston Gilcrease, Simone Baldovino, Andrea Ferreira Poshar, Alessio Conti, Roberta Fenoglio

**Affiliations:** 1University Center of Excellence on Nephrologic, Rheumatologic and Rare Diseases (ERK-Net, ERN-Reconnect and RITA-ERN Member) with Nephrology and Dialysis Unit and Center of Immuno-Rheumatology and Rare Diseases (CMID), Coordinating Center of the Interregional Network for Rare Diseases of Piedmont and Aosta Valley, San Giovanni Bosco Hub Hospital ASL Città di Torino and Department of Clinical and Biological Sciences, 10154 Turin, Italy; savino.sciascia@unito.it (S.S.); gabriele.ferrara606@edu.unito.it (G.F.); lorenzo.roccatello@gmail.com (L.R.); elena.rubini@unito.it (E.R.); silviagrazietta.foddai@unito.it (S.G.F.); massimo.radin@unito.it (M.R.); irene.cecchi@unito.it (I.C.); daniela.rossi@unito.it (D.R.); alice.barinotti@unito.it (A.B.); simone.baldovino@unito.it (S.B.); andrea.poshar@gmail.com (A.F.P.); roberta.fenoglio@unito.it (R.F.); 2Centre for Biostatistics, Epidemiology, and Public Health (C-BEPH), Department of Clinical and Biological Sciences, University of Turin, 10124 Turin, Italy; fulvio.ricceri@unito.it (F.R.); gregorywinston.wilcrease@unito.it (W.G.)

**Keywords:** systemic lupus erythematosus, diet, supplements, patient-driven research

## Abstract

Background/Objectives: Inflammation and immunological dysregulation are central to systemic lupus erythematosus (SLE), a complex autoimmune disease. Recently, there has been increasing interest in the potential role of dietary factors in SLE. This study aimed to explore the relationship between diet and SLE by addressing patient needs, conducting a systematic review, and providing guidance to the patient community. Methods: This four-step study started with a survey of patients with SLE that was conducted to gather frequently asked questions (FAQs) related to diet. Using the PICO framework, two comprehensive systematic literature searches were performed in PubMed to address these FAQs. Subsequently, the evidence retrieved was used to answer FAQs and provide guidance to people with SLE. A second survey was conducted to gather patient feedback on the topics and guidance provided. Results: A literature review of 28 systematic reviews was performed, evaluating the impact of diet on inflammation, immune response, and health outcomes in SLE patients. The review focused on key nutritional elements, including vitamin D, omega-3 fatty acids, curcumin supplements, and low-calorie or low-glycemic index diets. Seven guidance statements were developed based on these findings. All the answers provided were positively assessed by participants. Conclusions: This patient-centered study improves our understanding of the diet–SLE relationship through systematic reviews and patient feedback. While specific dietary recommendations for SLE are not yet established, patient input underscores the need for ongoing research to optimize treatment strategies and quality of life for those with SLE.

## 1. Introduction

Globally, around 5 million individuals suffer from systemic lupus erythematosus (SLE), a chronic and multifaceted autoimmune disease marked by inflammation and tissue damage [1]. SLE can affect individuals of any age, ethnicity, and gender, yet over 90% of newly diagnosed cases occur in women of childbearing age [2]. This autoimmune disease is characterized by the presence of autoantibodies against nuclear antigens, immune complex deposition, and chronic inflammation in organs like the skin, joints, and kidneys [1].

Although the pathophysiology and underlying causes of SLE remain unclear, research indicates that dietary, hormonal, genetic, and environmental variables can play an important role [3]. While multiple genes are known to contribute to SLE susceptibility [4], influencing immune regulatory mechanisms and autoantibody production, environmental agents, such as cigarette smoke or ultraviolet light [5,6], and hormonal influences can trigger and modulate disease activity [4]. Dietary factors, and especially their impact on the gut microbiota and epigenetic modifications, have been shown to contribute to the heterogeneous nature of SLE [7,8]. Therefore, to improve the management and prognosis of SLE, it is imperative to identify modifiable behaviors that may be more closely related to the pathogenesis of this disease and, accordingly, could potentially be used to control disease activity.

Preventing flares and organ damage, promoting long-term survival, and improving quality of life are the main goals of SLE management today [1]. Pharmacological therapy usually involves the long-term administration of glucocorticoids, immunosuppressive medications, and antimalarials [9]. This medication regimen can cause side effects with different degrees of severity, such as nausea and gastrointestinal issues, acne, and infections [10,11]. Despite advances in diagnosis and therapy, SLE remains a difficult disease to manage, and many of its patients carry a heavy physical and psychological burden [12,13]. In particular, some patients have been shown to benefit from alternative therapies, including diet interventions and lifestyle modifications, which may represent an additional resource for managing this difficult and severe condition [3].

Although pharmacological therapy remains the cornerstone of treatment, complementary interventions such as diet modification and nutritional supplementation may provide patients with additional resources to improve their health and quality of life [14]. In fact, the management and treatment of SLE should involve a combination of pharmacological therapy with nutritional and dietary interventions [15]. Studies have explored the role of nutrients such as vitamin D, omega-3 fatty acids, and curcumin in modulating chronic inflammation and immune dysregulation. Furthermore, low-calorie and low-glycemic index diets have been shown to reduce inflammation and improve metabolic markers in other chronic diseases [16,17]. However, even if people with SLE often ask healthcare professionals (HCPs) about the role of nutrition and its impact on their condition among the many potential factors involved, their knowledge of nutritional status and food intake are often inadequate [18]. Additionally, HCPs could be hampered in providing reliable information to people with SLE due to the lack of specific nutrition guidelines for this population.

Providing patients with scientifically reliable and trustworthy information is crucial in today’s information-driven environment, particularly when it comes to chronic conditions such as SLE [19]. Here, the HCPs can play a key role. Patients often seek HCPs’ guidance on what foods to avoid, what diets to follow, and what supplements to take [20]. By helping patients make decisions about their health and well-being using reliable and accurate information, HCPs can reduce the risk of complications and improve the quality of life of their patients [21]. Patient-driven research, in which patients actively participate in the research process, is increasingly acknowledged for its potential to generate reliable and relevant results [22]. This approach benefits from the unique insights and experiences of patients to shape research priorities, design, and implementation, ensuring that the studies address the real-world needs and concerns of the patient community [22,23].

This multistep, patient-driven study aimed to develop a list of frequently asked questions (FAQs) about food and dietary supplements among people with SLE and relative answers, producing a list of recommendations. The main objective of this study is to help patients make evidence-based decisions that will improve their overall health and well-being by educating them about how food and dietary supplements affect their health. Similarly, these FAQs can serve as a valuable resource for HCPs to identify and address the specific nutritional needs of people with SLE. By using these FAQs to guide patient conversations, HCPs can promote healthier lifestyle choices and improve overall health outcomes for this population.

## 2. Materials and Methods

### 2.1. Design

This study applied a four-step design (Figure 1) to first identify the unmet nutritional needs of people with SLE. Subsequently, a series of frequently asked questions (FAQs) were formulated and answered based on available evidence in the literature. Finally, to ensure consistency and credibility in the patient-driven procedure [22], such answers were evaluated in terms of satisfaction by the same people with SLE who expressed their unmet needs were evaluated in terms of satisfaction. The four steps are presented in detail below.

### 2.2. Inclusion and Exclusion Criteria

Participants in this study were included from the outpatient clinics of the University Center of Excellence on Nephrologic, Rheumatologic and Rare Diseases (ERK-Net, ERN-Reconnect and RITA-ERN Member) with the Nephrology and Dialysis Unit and Center of Immuno-Rheumatology and Rare Diseases (CMID). All consecutive patients with SLE who presented to the outpatient clinic on an index week (from November 6 to 11, 2023) were invited to participate in the study via direct contact with the researchers. The study was promoted during routine visits to encourage participation. To be included in the study, participants had to be 18 years or older, diagnosed with SLE for at least two years, and undergoing pharmacological treatment with immunosuppressive medications, glucocorticoids, and/or antimalarials, either alone or in combination [24]. People with SLE who could not understand the Italian language or who were attending a follow-up with a dietitian/nutritionist were excluded.

### 2.3. Steps


Step I: Identifying unmet needs (FAQs)


To initiate this study, we first identified the most common concerns among people with SLE about diet and nutritional supplements. An anonymous survey was designed to efficiently collect the unmet needs of people with SLE by encouraging honest and candid responses in a cost-effective manner. To this end, an open-ended question was crafted to gather the maximum amount of information from the participants while ensuring their clear responses. In the open question, participants were asked to provide their personal views or primary concerns about the relationship between diet and SLE by filling in ten lines of text.

A multidisciplinary panel of 10 HCPs, including physicians, dietitians, researchers, laypersons, and an expert patient, reviewed and approved the open-ended questions for relevance, clarity, and comprehensibility before the administration to participants.

The administration of the open-ended questions was performed using SurveyMonkey. Moreover, sociodemographic and clinical data were collected at the beginning of the survey. Measures were in place to prevent multiple submissions by the same participant. Additionally, a hard-copy, paper-and-pencil option was made available for those who preferred this type of compilation mode.

At the end of this step, a series of FAQs were formulated that were derived directly from the lived experiences of people with SLE, in accordance with the principles of patient-driven research [21].


Step II: Literature search


In this step, the previously formulated FAQs were converted into a corresponding research question using the PICO (population, intervention, comparison, and outcome) format. This approach optimized the search strategy by providing a structured framework and search strings for identifying relevant studies in the electronic database.

The PubMed database was used for the purpose of identifying all relevant studies conducted for each of the previously constructed PICOs. After an initial database search, the PICOs and search terms were refined with the help of an expert librarian, and a second database search was conducted. In both rounds, the records retrieved for each of the search strings were exported and uploaded into software for organizing bibliographic resources (Rayyan). Subsequently, the duplicate records were removed, and titles and abstracts were independently screened by two researchers (GF and SGF) to include exclusively the systematic reviews (SRs) conducted for each PICO. The exclusive inclusion of SRs was chosen to guarantee the highest quality of evidence in order to answer the FAQs formulated by people affected by SLE. Thus, primary studies or non-systematic secondary studies were excluded. Any disagreement regarding the eligibility of the SRs was resolved by achieving an agreement with a third researcher (SS).

For each of the included SRs, two independent researchers (ER and MR) extracted data on the review characteristics (author, country, and year), the number of studies included (pre-clinical and clinical), the inclusion criteria, and the main results. To collect the necessary data, a customized Excel spreadsheet was developed and utilized.


Step III: Guidance statements


After identifying the most relevant scientific evidence to address the FAQs formulated by people with SLE about diet and nutritional supplements, recommendations were developed to answer each FAQ. To achieve this, the fundamental techniques for evaluating and synthesizing the available data commonly used in the traditional guideline development processes were applied [25]. In brief, we developed this guideline through an iterative process informed by interviews, feedback, and a consensus process with a multidisciplinary panel, as previously described [25].


Step IV: Patients’ perception of retrieved evidence


The last step of this study consisted of the completion of a follow-up questionnaire by the same participants who took part in the first step. This tool was designed to investigate the subjective feedback of people with SLE about the answers to FAQs previously developed. To evaluate participant satisfaction with the developed recommendations, these were incorporated into the questionnaire as a Likert scale (strongly disagree—1, disagree—2, undecided—3, agree—4, and strongly agree—5). The use of a Likert scale is based on the belief that attitudes are measurable and that their strength or intensity can be linearly arrayed on a continuum from strong agreement to strong disagreement. The survey was administered through SurveyMonkey to all individuals who had participated in the first phase of the study.

### 2.4. Data Analysis

Data were analyzed through a process mirroring the steps described above.

The responses to the open-ended questions provided by the participants were systematized and coded through an iterative process, as described by Braun and Clarke [26]. Following this six-step thematic analysis approach, common themes were identified, starting from the responses provided by the participants. This provided a summary of the unmet needs of people with SLE in terms of diet and nutrition. The first step implied data familiarization by the researchers, who thoroughly read the transcriptions of the responses provided by the participants. Subsequently, two independent researchers (S.S. and S.G.F.) inductively coded the data, developing a coding framework, and the assigned codes were reviewed and validated by a third researcher (D.R.). Any discrepancies in the coding were discussed and resolved by refining the coding framework. Together, the previously involved researchers identified recurring patterns within the data, assigning specific second-level codes to these patterns. These second-level codes were then organized into themes and, where applicable, further subdivided into subthemes, based on their relatedness and parallel findings. To formulate more cohesive themes, second-level codes that shared similar content were grouped together. This grouping was subjected to repeated reviews to confirm its continued relevance to the overarching theme. Finally, the themes, subthemes, and codes were reviewed by the entire research group, which comprised both HCPs and an expert patient. This process involved examining the entire dataset to ensure that the analysis remained aligned with the study’s objectives. The themes, subthemes, and codes deemed of low importance were discarded. Meanwhile, related or overlapping themes and subthemes were merged to streamline the analysis. At the end of this process, each theme constituted an FAQ.

Findings extracted by the researchers that were contained in the SRs were synthesized narratively to present the available evidence for each of the identified FAQs.

The sociodemographic, clinical, and satisfaction data collected from SurveyMonkey were analyzed using GraphPad Prism 8 and Microsoft Excel 2010. The quantitative data analysis was descriptive, using mean and minimum–maximum ranges, while qualitative variables were presented as frequencies and percentages.

### 2.5. Ethical Considerations

The study was approved by the Territorial Ethics Committee of the Città della Salute e della Scienza di Torino University Hospital (protocol no. 0126695 of 31/10/2023). Participants received verbal and written information about the study. Participation was voluntary, and people with SLE agreed to complete the survey by signing an informed consent form. Participants could withdraw from the study at any time and for any reason without consequences. Data were anonymized during transcription, analysis, and presentation by assigning participants an alphanumerical code, and the results were presented as aggregated.

## 3. Results

### 3.1. Characteristics of Participants

Of the 30 patients attending the outpatient clinics during the study period, 22 responded to the survey, representing an overall response rate of 73.3%. Their main characteristics are described in Table 1.

### 3.2. Identifying Unmet Needs (FAQs)

The open-ended answers provided by people with SLE included an average of 9.2 lines (SD ± 0.8) of free text. Participants used a mean of 135 words (minimum of 107–maximum of 152) to describe their experiences and concerns about the relationship between their condition and nutrition.

The texts written by the participants produced 74 primary codes, which were later condensed into 19 **s**-level codes, leading to the identification of 4 main themes. These included the inflammatory properties of foods and their ability to trigger flares, possible supplements that can be taken, interferences between foods and treatments, and different types of diet available for people with SLE. In translating these themes into FAQs, two were subdivided into five more specific questions. This provided people with SLE with more focused guidance on their dietary behaviors. The final list of FAQs is presented in Table 2.

### 3.3. Literature Search

An initial PubMed search identified 69 records. After inclusion and exclusion criteria were applied, 49 records remained. The removal of duplicates and a review of these records resulted in the selection of 28 SRs in our study. Figure 2 provides a summary of the literature review process, including the search strategy and the articles that were found.

For the first FAQ, which was focused on the anti-inflammatory properties of foods, a total of 48 records were retrieved (Table 3). More information is provided in the Appendix A.

For the second FAQ, which was focused on the possibilities for a dietary regimen reducing or avoiding flares, a total of 3 records were retrieved (Table 4). More information is provided in the Appendix A.

For the third, fourth, and fifth FAQs, which were focused on the supplements, a total of 27 records were retrieved (Table 5). More information is provided in the Appendix A.

For the sixth FAQ, which was focused on potential interferences between foods and therapies, a summary of results is summarized in Table 6. More information is provided in the Appendix A.

For the seventh FAQ, which was focused on benefits deriving from a low-calorie or low-glycemic diet, a total of 5 records were retrieved (Table 7). More information is provided in the Appendix A.

For each of the FAQs, the findings of SRs were summarized, and for each FAQ, a statement was developed and refined by the researchers.

### 3.4. Guidance Statements

Following a thorough examination of the main results obtained from previously identified SRs, a technical and social process was used to generate the suggestions meant to address the FAQs. While the social phase involved creating statements that addressed unmet needs and would be feasible for individuals with SLE, the technical approach involved synthesizing evidence from the previously outlined literature searches. Leveraging the previously outlined methodology, 7 statements were developed, answering the FAQs (Table 8), which were subsequently subjected to the assessment by the same people with SLE who participated in the first step.

### 3.5. Patients’ Perception of Retrieved Evidence

All the participants who took part in the first step of this study answered the follow-up survey (response rate 100%). All the answers provided to the FAQs, which were developed through an evidence-to-guidance process, received a score that was consistently at or above three out of five, denoting a positive assessment by people with SLE (as shown in Figure 3). In particular, the higher level of agreement expressed by participants was reported in statements 3, 4, and 6.

## 4. Discussion

The purpose of this multistep, patient-driven study was to develop a list of FAQs about food and dietary supplements among people with SLE, which were subsequently answered providing a recommendation to each of them. The unmet needs of participants included the possibility that foods can cause flares or produce inflammation, concerns about nutritional supplementation, the potential effect of some foods on the efficacy of therapy, and available diets for people with lupus. The SRs identified through the literature review process highlighted some viable answers to the FAQs, which could help people with SLE make evidence-based decisions and support HCPs in providing targeted education about nutrition.

Understandably, many people with SLE are curious about whether changing their diet or hydration habits could help manage their disease. An example would be the first two FAQs, which focused on the anti-inflammatory properties of foods and the effect of diet on reducing flares. However, identifying the precise relationship between nutrition and lupus is challenging. Research on how diet affects SLE patients is complex, rarely identifying specific foods to take or avoid, given the complexity of SLE. This may lead to patients’ unwillingness to change their dietary habits, even though a previous study showed that all people with SLE would be likely to change their diet if it helped decrease their disease symptoms [52].

Oftentimes, scientists and physicians make specific dietary recommendations based on studies involving SLE-affected mice, though these do not directly translate to human nutrition studies [53]. Although direct clinical research on nutrition’s impact on SLE is limited, large-scale studies across various cultures have shown a link between certain dietary patterns and a reduced risk of inflammation and some chronic diseases [54]. An anti-inflammatory diet rich in fruits, vegetables, whole grains (such as barley, quinoa, and wild rice), legumes, and nuts is particularly beneficial for health. Furthermore, extensive evidence suggests that minimal consumption of processed foods correlates with better health outcomes. The foundation of an anti-inflammatory diet is generally agreed to be high in fruits, vegetables, and plant-based proteins, despite some disagreements over specifics.

While such a diet may not cure any chronic inflammatory condition, it could enhance overall well-being and alleviate some SLE symptoms. Diets known for their anti-inflammatory properties, such as the Mediterranean diet, have been associated with lower levels of inflammation and chronic disease [55]. This diet’s guidelines can be adapted by almost any culture, with the Mediterranean region itself sharing a common dietary pattern that includes moderate amounts of fish, poultry, low-fat dairy, and nuts, along with high intakes of fruits, vegetables, olive oil, whole grains, and legumes. It advises less consumption of red meat, processed meats, sugary drinks (including fruit juice), salt, and processed foods.

On the other hand, there is evidence that the dietary intake of unsaturated fatty acids may play a role in reducing SLE activity and may significantly influence HDL levels without significantly influencing inflammation markers or other lipid profiles [53]. Similarly, vitamin D and E intake, as well as curcumin, and its preventive action on inflammatory markers, may be a resource for people with SLE [16,17]. The therapeutic potential of nutritional approaches that could modulate the development of major comorbidities related to cardiovascular disease risk has been demonstrated in patients with SLE, including the cessation of smoking and the adoption of healthy lifestyles, including appropriate nutrition, has been demonstrated [3]. Furthermore, the potential efficacy of polyphenols in enhancing endothelial function in individuals with chronic diseases should be considered when recommending dietary regimens for SLE [56,57]. All of these recommendations should be evaluated by HCPs and discussed with people with SLE, given their perceived lack of support regarding nutrition during clinical consultations, as well as their willingness to change their diet if effectiveness in controlling disease progress is demonstrated [52].

Our study explores the impact of nutrition on people with SLE, highlighting the lack of evidence-based recommendations for personalized diet changes. This is also reflected in the absence of any specific recommendation on nutrition within non-pharmacological treatment guidelines for people with SLE, which, among healthy lifestyles, neglect diet in particular [58]. Drawing on biological research, our conclusions consider various perspectives, including Western medicine, traditional healing practices, and cultural beliefs, as well as research on SLE in mice. From an exploratory study, we compiled a list of seven general statements to address common questions from SLE patients.

Overall, no specific diet or nutritional plan has been consistently proven to be effective in managing disease activity in SLE patients. For example, Gavilan-Carrera et al. showed that a 12-week aerobic training intervention did not improve either clinical indexes or adherence to the Mediterranean diet in a sample of women with SLE, suggesting that a dietary intervention could be more effective than a physical intervention [59]. Our research also points to the difficulty of finding public data to guide personalized dietary decisions for patients. Moreover, several methodological considerations should be considered. According to the study, where evidence summaries for different management approaches are comparable, evidence-based recommendations should be both localized and globalized [60]. The vast variability in the research we found throughout our search limited the comparing procedure in certain ways. Second, several reasons exist as to why the degree of confidence in effect estimates (i.e., quality of evidence) could differ. Variations in the relative values of good and bad outcomes might primarily lead to different tolerance limits [61].

To improve the design of clinical trials and assess the viability of treatment interventions centered around diet, it is essential for future studies to concentrate on patient conversations.

The limitations of this study include a possible selection bias among participants who may have been more interested in completing the survey based on their prior interest in the research topic. Furthermore, the choice to recruit people with SLE during a one-week index period could have limited the generalizability of the included participants. However, the sample is consistent with the clinical and sociodemographic characteristics of people with SLE, and it is representative of this population. Furthermore, the application of a multi-phase approach in which people with SLE were included in each step of the study minimized the distance between researchers and stakeholders, ensuring credibility and trustworthiness of the findings obtained and improving the potential for their translation into the daily clinical practice.

## 5. Conclusions

Several studies have explored the role of nutrition and diet in addressing the immunological dysregulation and chronic inflammation of SLE. While findings are promising, the available evidence lacks consistency, making it difficult to pinpoint specific foods, diets, or supplements that are effective in managing SLE. No single dietary approach has been proven to consistently influence disease activity. However, positive feedback from patients and caregivers highlights the need for further investigation into the impact of nutrition on SLE-related symptoms. Future interventional studies that account for socioeconomic and clinical characteristics are essential to identify dietary strategies that could improve SLE management.

## Figures and Tables

**Figure 1 nutrients-16-04132-f001:**
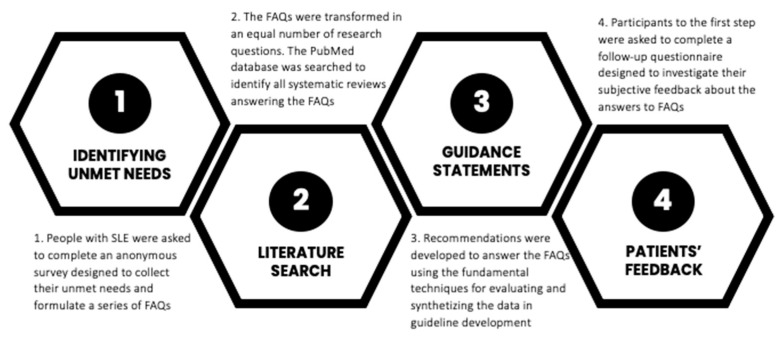
Visual representation of the four-step study design.

**Figure 2 nutrients-16-04132-f002:**
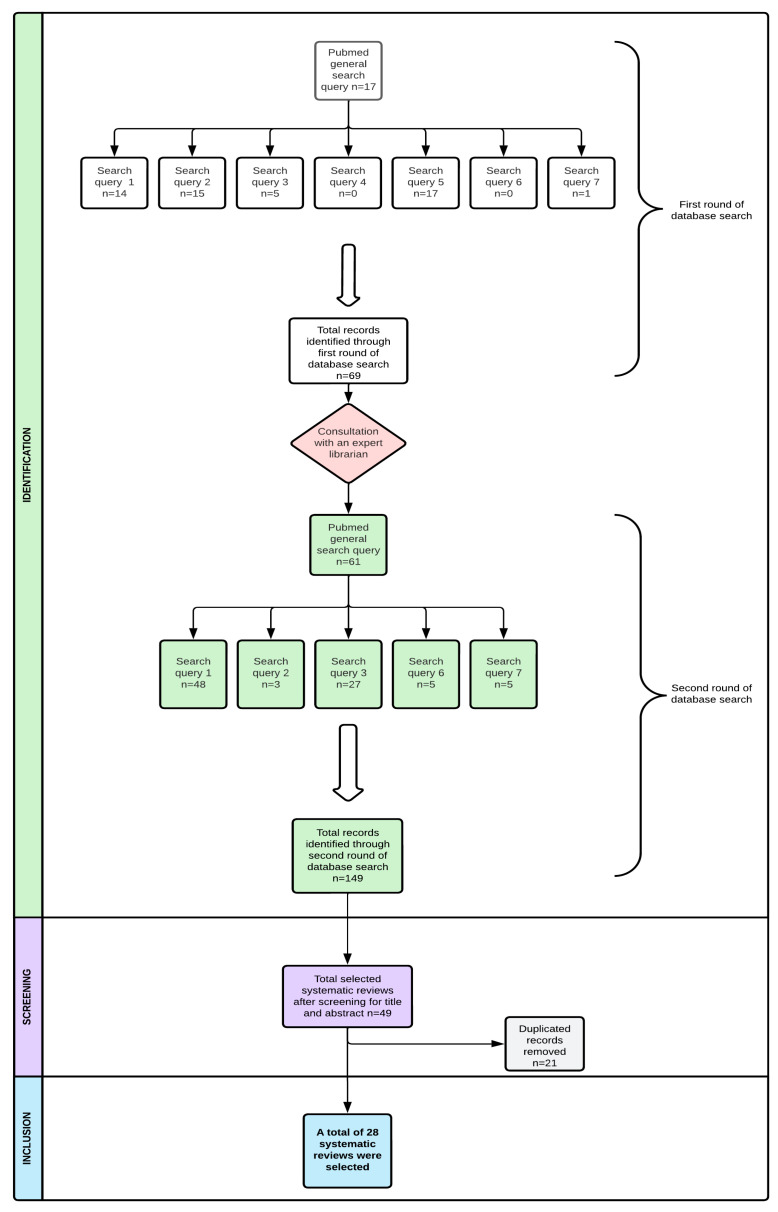
Literature review process.

**Figure 3 nutrients-16-04132-f003:**
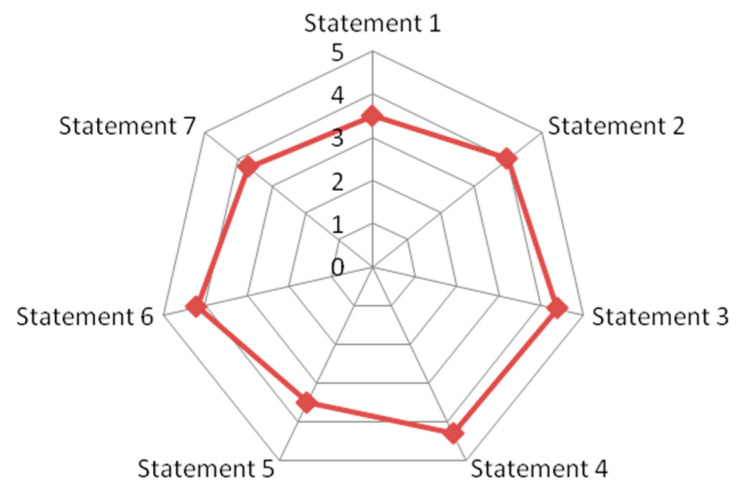
Spider chart summarizing the Likert ratings for all the proposed statements.

**Table 1 nutrients-16-04132-t001:** Characteristics of participants.

Demographic and Clinical Characteristics	Participants (n = 22)
Age, years (median, min–max range)	34 (22–56)
Years of follow-up (median, min–max range)	7 (2–11)
Female *N*. (%)	21 (95)
Articular involvement *N*. (%)	20 (90)
Skin involvement *N*. (%)	14 (63)
Renal involvement *N*. (%)	12 (54)
Hematological involvement *N*. (%)	11 (50)

**Table 2 nutrients-16-04132-t002:** Frequently asked questions about dietary and food supplements among people with systemic lupus erythematosus.

**1.**	Are there any anti-inflammatory foods I can benefit from?
**2.**	What kind of diet can I follow to avoid/reduce flare-ups?
**3.**	Are there any supplements that I can benefit from?
**4.**	Can I take protein supplements?
**5.**	Can I take vitamin supplements?
**6.**	Are there any foods that can interfere with my therapy?
**7.**	Can I benefit from a low-calorie or low-glycemic diet?

**Table 3 nutrients-16-04132-t003:** Included studies in the systematic review examining the anti-inflammatory properties of foods.

Authors (Year)	Study	# Studies Included	Main Results
Antico A, et al. (2012)	Can supplementation with vitamin D reduce the risk or modify the course of autoimmune diseases? A systematic review of the literature [27]	219	Experimental studies of humans suggest that vitamin D supplementation exerts beneficial effects in reducing disease severity. However, the supplied doses may be insufficient to control autoimmune conditions or prevent their onset. The existing literature does not provide enough evidence to establish a direct link between vitamin D deficiency and the incidence of autoimmune diseases.
de Medeiros M, et al. (2019)	Dietary intervention and health in patients with systemic lupus erythematosus: A systematic review of the evidence [28]	11	Omega-3 supplementation reduces inflammation, disease activity, endothelial dysfunction, and oxidative stress. Vitamin D supplementation raises serum levels and decreases inflammatory and hemostatic markers. Turmeric supplementation reduces proteinuria, hematuria, and systolic blood pressure, while a low-glycemic index diet promotes weight loss and decreases fatigue.
Duarte-García A, et al. (2020)	Effect of omega-3 fatty acids on systemic lupus erythematosus disease activity: A systematic review and meta-analysis [29]	8	Omega-3 fatty acids reduced disease activity when compared to placebo. This meta-analysis suggests that omega-3 fatty acids could provide a therapeutic benefit in addition to immunosuppressive regimens used for SLE.
Franco A, et al. (2017)	Vitamin D supplementation and disease activity in patients with immune-mediated rheumatic diseases [30]	9 (3 on SLE)	Vitamin D supplementation reduced anti-dsDNA positivity on SLE. Vitamin D supplementation may be beneficial to patients with high anti-dsDNA positivity, possibly reducing clinical flares.
Guan S, et al. (2019)	Association between circulating 25- hydroxyvitamin D and systemic lupus erythematosus: A systematic review and meta-analysis [31]	19	Regardless of age, disease duration, or therapy (whether corticosteroids, immunosuppressive drugs, or neither), circulating 25(OH)D levels were significantly reduced in SLE patients. Vitamin D deficiency significantly increases SLE risk, while insufficiency slightly decreases it (not significantly), and sufficiency significantly decreases the risk.
Gwinnutt J, et al. (2022)	Effects of diet on the outcomes of rheumatic and musculoskeletal diseases (RMDs): Systematic review and meta-analyses informing the 2021 EULAR recommendations for lifestyle improvements in people with RMDs [32]	174 (24 SRs, 150 OAs)	High consumption of vitamin B6, fiber, and vitamin C was associated with a lower risk of developing an active disease. Two out of three studies included in the systematic review reported reductions in disease activity following omega-3 intervention. The evidence for fish oil/omega-3 for SLE was rated as moderate but showed no effect on outcomes.
Irfan S, et al. (2022)	Effects of Vitamin D on Systemic Lupus Erythematosus Disease Activity and Autoimmunity: A Systematic Review and Meta-Analysis [33]	6	In SLE patients, vitamin D supplementation significantly decreased disease activity scores and increased C3 levels. Its effect on fatigue was inconclusive, and no significant changes were observed in anti-dsDNA and C4 levels.
Islam M, et al. (2020)	Immunomodulatory Effects of Diet and Nutrients in Systemic Lupus Erythematosus (SLE): A Systematic Review [34]	184	A low-calorie, low-protein diet rich in fiber, polyunsaturated fatty acids, vitamins, minerals, and polyphenols contains sufficient macronutrients and micronutrients to regulate disease activity in SLE by modulating inflammation and immune functions.
Islam M, et al. (2019)	Vitamin D status in patients with systemic lupus erythematosus (SLE): A systematic review and meta-analysis [35]	34	Serum vitamin D levels were found to be significantly lower in SLE patients compared to healthy controls.
Jiao H, et al. (2022)	Diet and Systemic Lupus Erythematosus (SLE): From Supplementation to Intervention [15]	14	Vitamin D or E supplementation improved inflammatory markers or antibody production but did not affect disease activity scores. The addition of curcumin to vitamin D showed no added benefit. Omega-3 supplementation reduced ESR, CRP, disease activity, inflammatory markers, oxidative stress, and improved lipid levels and endothelial function. A low-glycemic index diet helped reduce weight and improve fatigue.
Marton L, et al. (2022)	Curcumin, autoimmune and inflammatory diseases: going beyond conventional therapy–a systematic review [36]	36 (2 on SLE)	Turmeric supplementation significantly reduced proteinuria, systolic pressure, and hematuria in lupus nephritis patients without any adverse effects. The short-term use of turmeric may serve as a safe adjuvant therapy. However, no significant changes were observed in disease activity scores, IL-6, or TGF-β levels. Beneficial doses of curcumin ranged from 20 to 500 mg orally.
Ramessar N, et al. (2023)	The impact of curcumin supplementation on systemic lupus erythematosus and lupus nephritis: A systematic review [37]	13	In human trials, curcumin reduced 24-h and spot proteinuria, but the trials were small (14–39 patients) with varying doses and durations (4–12 weeks). No significant changes were observed in C3, dsDNA, or disease activity scores. In mouse models, curcumin (1 mg/kg/day for 14 weeks) suppressed NF-κβ activation and NOS expression, leading to reductions in dsDNA, proteinuria, renal inflammation, and IgG subclasses, along with decreases in Th1/Th17 cells, IL-6, and ANA levels.
Ramessar N, et al. (2022)	The effect of Omega-3 fatty acid supplementation in systemic lupus erythematosus patients: A systematic review [38]	13	Current data indicate potential benefits for disease activity, as shown by improvements in the disease activity scores, along with changes in plasma-membrane arachidonic acid composition and reduced urinary 8-isoprostane levels, with minimal adverse events.
Ravi N, et al. (2022)	The Key Role of Glutathione Compared to Curcumin in the Management of Systemic Lupus Erythematosus: A Systematic Review [39]	15	Curcumin and glutathione are potential treatments for SLE, with curcumin being a more promising alternative due to its action on multiple pathways and its greater accessibility.
Sahebari M, et al. (2014)	Correlation between serum 25(OH)D values and lupus disease activity: An original article and a systematic review with meta-analysis focusing on serum Vit D confounders [40]	38 (11 in the meta-analysis)	The results of this meta-analysis show an inverse correlation between vitamin D levels and disease activity in SLE.
Sakthiswary R, Raymond A. (2013)	The Clinical Significance of Vitamin D in Systemic Lupus Erythematosus: A Systematic Review [41]	22	There is strong evidence of a statistically significant inverse relationship between vitamin D levels and SLE disease activity.
Salman-Monte T, et al. (2017)	Bone mineral density and vitamin D status in systemic lupus erythematosus (SLE): A systematic review [42]	Ns	SLE patients are at risk for low vitamin D levels and reduced bone mineral density, making it essential to study, monitor, prevent, and treat bone metabolism disorders in this population.
Sousa J, et al. (2017)	Effect of vitamin D supplementation on patients with systemic lupus erythematosus: a systematic review [43]	4	Three studies showed that vitamin D supplementation reduced disease activity and improved inflammatory markers, fatigue, and endothelial function in SLE patients. This review supports the benefits of vitamin D supplementation for those with vitamin D insufficiency or deficiency.
Tsai T, et al. (2021)	Serum Homocysteine, Folate, and Vitamin B12 Levels in Patients with Systemic Lupus Erythematosus: A Meta-Analysis and Meta-Regression [44]	50	SLE patients had higher serum homocysteine and lower vitamin B12 levels compared to individuals without SLE. Meta-regression analysis showed an inverse correlation between homocysteine and C3/C4 levels. There is currently insufficient evidence to support vitamin B12 and/or folate supplementation for these patients.
Zeng L, et al. (2022)	Curcumin and Curcuma longa Extract in the Treatment of 10 Types of Autoimmune Diseases: A Systematic Review and Meta-Analysis of 31 Randomized Controlled Trials [45]	34 (2 on SLE)	Curcumin may enhance regulatory T-cell responses by inhibiting antibody–antigen interactions, reducing tissue deposition, and limiting antibody production. However, more randomized controlled trials are needed to confirm its therapeutic effect and safety in SLE.

Legend: SLE, systemic lupus erythematosus; SR, systematic review; OA, original article; Ns, not specified.

**Table 4 nutrients-16-04132-t004:** Included studies examining the possibilities for a dietary regimen reducing/avoiding flares.

Authors (Year)	Study	# Studies Included	Main Results
Islam M, et al. (2020)	Immunomodulatory Effects of Diet and Nutrients in Systemic Lupus Erythematosus (SLE): A Systematic Review [34]	184	A low-calorie, low-protein diet rich in fiber, polyunsaturated fatty acids, vitamins, minerals, and polyphenols contains sufficient macronutrients and micronutrients to regulate disease activity in SLE by modulating inflammation and immune functions.
Rodríguez Huerta M, et al. (2016)	Healthy lifestyle habits for patients with systemic lupus erythematosus: A systemic review [46]	21	Smoking increases the risk of skin damage and disease activity in SLE patients. A diet rich in polyunsaturated fatty acids, regular exercise, and avoiding a sedentary lifestyle should be recommended for patients with stable SLE
Wieczorek M, et al. (2022)	Smoking, alcohol consumption and disease-specific outcomes in rheumatic and musculo-skeletal diseases (RMDs): systematic reviews informing the 2021 EULAR recommendations for life-style improvements in people with RMDs [47]	90	SLE patients who smoke tend to have worse outcomes, including lower mental and physical quality of life scores, more rashes, higher disease activity, and increased cardiovascular morbidity.

Legend: SLE, systemic lupus erythematosus; SR, systematic review; OA, original article; Ns, not specified.

**Table 5 nutrients-16-04132-t005:** Included studies examining the supplements.

Authors (Year)	Study	# Studies Included	Main Results
Antico A, et al. (2012)	Can supplementation with vitamin D reduce the risk or modify the course of autoimmune diseases? A systematic review of the literature [27]	219	Experimental studies in humans suggest that vitamin D supplementation exerts beneficial effects in reducing disease severity. However, the supplied doses may be insufficient to control autoimmune conditions or prevent their onset. The existing literature does not provide enough evidence to establish a direct link between vitamin D deficiency and the incidence of autoimmune diseases.
de Medeiros M, et al. (2019)	Dietary intervention and health in patients with systemic lupus erythematosus: A systematic review of the evidence [28]	11	Omega-3 supplementation reduces inflammation, disease activity, endothelial dysfunction, and oxidative stress. Vitamin D supplementation raises serum levels and decreases inflammatory and hemostatic markers. Turmeric supplementation reduces proteinuria, hematuria, and systolic blood pressure, while a low-glycemic index diet promotes weight loss and decreases fatigue.
Duarte-García A, et al. (2020)	Effect of omega-3 fatty acids on systemic lupus erythematosus disease activity: A systematic review and meta-analysis [29]	8	Omega-3 fatty acids reduced disease activity when compared to placebo. This meta-analysis suggests that omega-3 fatty acids could provide a therapeutic benefit in addition to immunosuppressive regimens used for SLE.
Franco A, et al. (2017)	Vitamin D supplementation and disease activity in patients with immune-mediated rheumatic diseases [30]	9 (3 on SLE)	Vitamin D supplementation reduced anti-dsDNA positivity on SLE. Vitamin D supplementation may be beneficial for patients with high anti-dsDNA positivity, possibly reducing clinical flares.
Guan S, et al. (2019)	Association between circulating 25-hydroxyvitamin D and systemic lupus erythematosus: A systematic review and meta-analysis [31]	19	Regardless of age, disease duration, or therapy (whether corticosteroids, immunosuppressive drugs, or neither), circulating 25(OH)D levels were significantly reduced in SLE patients. Vitamin D deficiency significantly increases SLE risk, while insufficiency slightly decreases it (not significantly), and sufficiency significantly decreases the risk.
Islam M, et al. (2020)	Immunomodulatory Effects of Diet and Nutrients in Systemic Lupus Erythematosus (SLE): A Systematic Review [34]	184	A low-calorie, low-protein diet rich in fiber, polyunsaturated fatty acids, vitamins, minerals, and polyphenols contains sufficient macronutrients and micronutrients to regulate disease activity in SLE by modulating inflammation and immune functions.
Islam M, et al. (2019)	Vitamin D status in patients with systemic lupus erythematosus (SLE): A systematic review and meta-analysis [35]	34	Serum vitamin D levels were found to be significantly lower in SLE patients compared to healthy controls.
Jiao H, et al. (2022)	Diet and Systemic Lupus Erythematosus (SLE): From Supplementation to Intervention [15]	14	Vitamin D or E supplementation improved inflammatory markers or antibody production but did not affect disease activity scores. The addition of curcumin to vitamin D showed no added benefit. Omega-3 supplementation reduced ESR, CRP, disease activity, inflammatory markers, and oxidative stress and improved lipid levels and endothelial function. A low-glycemic index diet helped reduce weight and improve fatigue.
Ramessar N, et al. (2023)	The impact of curcumin supplementation on systemic lupus erythematosus and lupus nephritis: A systematic review [37]	13	In human trials, curcumin reduced 24-h and spot proteinuria, but the trials were small (14–39 patients) with varying doses and durations (4–12 weeks). No significant changes were observed in C3, dsDNA, or disease activity scores. In mouse models, curcumin (1 mg/kg/day for 14 weeks) suppressed NF-κβ activation and NOS expression, leading to reductions in dsDNA, proteinuria, renal inflammation, and IgG subclasses, along with decreases in Th1/Th17 cells, IL-6, and ANA levels.
Ramessar N, et al. (2022)	The effect of Omega-3 fatty acid supplementation in systemic lupus erythematosus patients: A systematic review [38]	13	Current data indicate potential benefits on disease activity, as shown by improvements in the disease activity scores, along with changes in plasma-membrane arachidonic acid composition and reduced urinary 8-isoprostane levels, with minimal adverse events.
Sahebari M, et al. (2014)	Correlation between serum 25(OH)D values and lupus disease activity: An original article and a systematic review with meta-analysis focusing on serum Vit D confounders [40]	38 (11 in the meta-analysis)	The results of this meta-analysis show an inverse correlation between vitamin D levels and disease activity in SLE.
Sousa J, et al. (2017)	Effect of vitamin D supplementation on patients with systemic lupus erythematosus: a systematic review [43]	4	Three studies showed that vitamin D supplementation reduced disease activity and improved inflammatory markers, fatigue, and endothelial function in SLE patients. This review supports the benefits of vitamin D supplementation for those with vitamin D insufficiency or deficiency.
Yuen H, Cunningham M. (2014)	Optimal management of fatigue in patients with systemic lupus erythematosus: A systematic review [48]	26	Nine strategies for to alleviating fatigue in SLE patients were identified. Aerobic exercise and belimumab show the strongest evidence of efficacy. N-acetylcysteine and ultraviolet-A1 phototherapy show low-to-moderate evidence, while psychosocial interventions, dietary changes, vitamin D supplementation, and acupuncture show weak evidence. Dehydroepiandrosterone is not recommended due to insufficient evidence.

Legend: SLE, systemic lupus erythematosus; SR, systematic review; OA, original article; Ns, not specified.

**Table 6 nutrients-16-04132-t006:** Included studies examining the potential interferences between foods and therapies.

Authors (Year)	Study	# Studies Included	Main Results
Tan C, Lee S. (2021)	Warfarin and food, herbal or dietary supplement inter-actions: A systematic review [49]	149	While most foods, herbs, and supplements are safe in moderation, healthcare professionals should be aware of the increased risk of bleeding associated with certain foods and herbs, including Chinese wolfberry, chamomile tea, cannabis, cranberry, chitosan, green tea, Ginkgo biloba, ginger, spinach, St. John’s Wort, sushi, and smoking tobacco.

Legend: SLE, systemic lupus erythematosus; SR, systematic review; OA, original article; Ns, not specified.

**Table 7 nutrients-16-04132-t007:** Included studies examining the benefits deriving from low-calorie or low-glycemic diet.

Authors (Year)	Study	# Studies Included	Main Results
Andrades C, et al. (2017)	Management of cardiovascular risk in systemic lupus erythematosus: A systematic review [50]	19	Low-calorie and/or low-glycemic index diets may help with secondary prevention in obese SLE patients, while exercise can improve endothelial function as measured via flow-mediated dilation.
Imoto A, et al. (2021)	The impact of a low-calorie, low-glycaemic diet on systemic lupus erythematosus: a systematic review [51]	3	The diet improved quality of life and may have clinical relevance for the lipid profile. However, it did not affect fatigue, sleep quality, or disease activity based on multidisciplinary evaluations. A low-glycemic index diet showed a favorable effect on fatigue.
Islam M, et al. (2020)	Immunomodulatory Effects of Diet and Nutrients in Systemic Lupus Erythematosus (SLE): A Systematic Review [34]	184	A low-calorie, low-protein diet rich in fiber, polyunsaturated fatty acids, vitamins, minerals, and polyphenols contains sufficient macronutrients and micronutrients to regulate disease activity in SLE by modulating inflammation and immune functions.
Yuen H, Cunningham M. (2014)	Optimal management of fatigue in patients with systemic lupus erythematosus: A systematic review [48]	26	Nine strategies for alleviating fatigue in SLE patients were identified. Aerobic exercise and belimumab show the strongest evidence of efficacy. N-acetylcysteine and ultraviolet-A1 phototherapy show low-to-moderate evidence, while psychosocial interventions, dietary changes, vitamin D supplementation, and acupuncture show weak evidence. Dehydroepiandrosterone is not recommended due to insufficient evidence.

Legend: SLE, systemic lupus erythematosus; SR, systematic review; OA, original article; Ns, not specified.

**Table 8 nutrients-16-04132-t008:** Statements answering the FAQs on dietary and food supplements among people with systemic lupus erythematosus.

**Statement 1**	Limited evidence suggests that supplements like vitamin D, curcumin, and omega-3 fatty acids may influence immunological inflammatory markers. However, further research is needed to clearly understand their direct impact on disease activity in SLE.
**Statement 2**	There is no consistent evidence to support that any specific diet or exercise regimen effectively reduces the activity of SLE.
**Statement 3**	It is generally recommended as good clinical practice to adhere to a balanced diet. Yet, no dietary supplement has conclusively been shown to worsen disease activity in SLE patients.
**Statement 4**	The specific effects of protein supplements, particularly those aimed at increasing lean body mass, on SLE have not been thoroughly investigated. Patients with renal issues are advised to proceed with caution.
**Statement 5**	Some studies suggest that vitamin D and, to a lesser extent, vitamin E supplements, might play a role in managing immunological inflammatory parameters. It is crucial that any supplementation is carried out under medical supervision.
**Statement 6**	Current research does not specifically address how individual foods directly impact SLE management. General insights can be drawn from evidence in the broader population, with notable interactions between vitamin K and certain anticoagulants, like Coumadin, and grapefruit’s effect on the bioavailability of specific drugs, such as mTOR inhibitors, serving as key examples.
**Statement 7**	In the absence of other conditions that may require a low-glycemic or low-protein diet, like diabetes or chronic kidney disease, there is considerable variability in the evidence supporting specific dietary approaches for SLE.

## Data Availability

The raw data supporting the conclusions of this article will be made available by the authors upon request. The data are not publicly available due to legal reasons.

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
