# Peer review of "The Interconnection Between Systemic Lupus Erythematosus and Diet: Unmet Needs, Available Evidence, and Guidance—A Patient-Driven, Multistep-Approach Study"

_nutrients, 2024, doi:10.3390/nu16234132_

Round 1
Reviewer 1 Report
Comments and Suggestions for Authors
The Authors aimed to analyze the relationship between diet and SLE taking into account patients’ needs, review of the available literature and guidance for the patients.
Considering complexity of the etiopathogenetic background of SLE the influence of dietary factors might be of significant importance so in this regard the study is fully justified.
The study is well designed and methods are correctly used.
There is a small sample of patients included to the study, but the analysis of dietary aspects is very detailed and allows the validation of the results.
The undoubted advantage is very thorough analysis of the literature.
For the future studies it would be good to include also other aspects e.g. socioeconomic, educational or specific manifestations like CVD or metabolic syndrome to check if there are some specific directions possible, other than for the general population, in terms of cardiovascular risk etc.
Author Response
Comments/suggestions by reviewer 1.
- The Authors aimed to analyze the relationship between diet and SLE taking into account patients’ needs, review of the available literature and guidance for the patients.
Considering complexity of the etiopathogenetic background of SLE the influence of dietary factors might be of significant importance so in this regard the study is fully justified.
The study is well designed, and methods are correctly used.
There is a small sample of patients included to the study, but the analysis of dietary aspects is very detailed and allows the validation of the results.
The undoubted advantage is very thorough analysis of the literature.
For the future studies it would be good to include also other aspects e.g. socioeconomic, educational or specific manifestations like CVD or metabolic syndrome to check if there are some specific directions possible, other than for the general population, in terms of cardiovascular risk etc.
Thank you for your feedback. We appreciated the positive reaction with which you acknowledged the results of our study. We have included a sentence in the conclusions addressing the point you have raised. Page 16, Lines 453-455: “Future interventional studies that account for socioeconomic and clinical characteristics are essential to identify dietary strategies that could improve SLE management.”
Reviewer 2 Report
Comments and Suggestions for Authors
First, I congratulate the authors for their wonderfully written work titled “The interconnection between systemic lupus erythematosus 2 and diet: unmet needs, available evidence, and guidance. A 3 patient-driven multistep approach study”. It is well-structured and covers important notions.
However, I have a few comments which if taken into consideration can increase the clarity and worthiness.
The authors should provide a flow chart of their material and method explaining the step-by-step process. the authors have mentioned studies related to smoking, drug consumption or alchohol how it can affect such chronic diseases like SLE, there are some study which can be compared in the discussion part emphasizing on polyphenols and their effects in chronic diseases. Badalica, M.; Munteanu, M.; Sturza, A.; Noveanu, L.; Streian, C.G.; Socaciu, C.; Muntean, D.; Timar, R.; Dragan, S. Characterization of the Effects of Two Polyphenols-Rich Plant Extracts on Isolated Diabetic Human Mammary Arteries. Revista De Chimie 2014, 65, 861-864. And doi:10.3390/pharmaceutics16091176. Furthermore, the authors should use the full form then abbreviate and later can use the abbreviations. But at least once you need to show what the abbreviation stands for, in their first use. similarly also in the abstract avoid using direct abbreviations. Grammatical errors require a native speaker. The conclusion requires reformulation of sentences and more direct approach. The clarity of figure 1 is very much poor, increase its resolution
Comments on the Quality of English Language
Minor grammatical issues need to be solved.
Author Response
Comments/suggestions by reviewer 2
- First, I congratulate the authors for their wonderfully written work titled “The interconnection between systemic lupus erythematosus and diet: unmet needs, available evidence, and guidance. A patient-driven multistep approach study”. It is well-structured and covers important notions. However, I have a few comments which if taken into consideration can increase the clarity and worthiness.
Thank you for your honest feedback. We have considered all your comments and provided a point-by-point reply. We hope these will improve the quality of our manuscript.
- The authors should provide a flow chart of their material and method explaining the step-by-step process.
Thank you for your suggestion. We have provided a figure (Page 3, Line 112 – Figure 1) to visually explain the four-step study design we applied.
- The authors have mentioned studies related to smoking, drug consumption or alchohol how it can affect such chronic diseases like SLE, there are some study which can be compared in the discussion part emphasizing on polyphenols and their effects in chronic diseases. Badalica, M.; Munteanu, M.; Sturza, A.; Noveanu, L.; Streian, C.G.; Socaciu, C.; Muntean, D.; Timar, R.; Dragan, S. Characterization of the Effects of Two Polyphenols-Rich Plant Extracts on Isolated Diabetic Human Mammary Arteries. Revista De Chimie 2014, 65, 861-864. And doi:10.3390/pharmaceutics16091176.
Thank you for your suggestion. We added a specific sentence in the discussion addressing this point. Page 16, Lines 401-403 “Furthermore, the potential efficacy of polyphenols in enhancing endothelial function in individuals with chronic diseases should be considered when recommending dietary regimens for SLE [with references 56 & 57 the studies you have kindly suggested].”
- Furthermore, the authors should use the full form then abbreviate and later can use the abbreviations. But at least once you need to show what the abbreviation stands for, in their first use. Similarly, also in the abstract avoid using direct abbreviations.
Thank you for your suggestion. We have now checked our manuscript and provided the full form before abbreviations in all the text. Regarding the use of abbreviations in the abstract, we used the abbreviations consistently with the journal’s instructions for Authors (https://www.mdpi.com/journal/nutrients/instructions - Acronyms/Abbreviations/Initialisms should be defined the first time they appear in each of three sections: the abstract; the main text; the first figure or table. When defined for the first time, the acronym/abbreviation/initialism should be added in parentheses after the written-out form).
- Grammatical errors require a native speaker.
Thank you for your suggestion. A native speaker who is an author of this study provided a grammatical check of the manuscript. We hope this new version is now more correct and understandable.
- The conclusion requires reformulation of sentences and more direct approach.
Thank you for your suggestion. We have rephrased all the sentences following your guidance. Page 17, Lines 445-455: “Several studies have explored the role of nutrition and diet in addressing the immunological dysregulation and chronic inflammation of SLE. While findings are promising, the available evidence lacks consistency, making it difficult to pinpoint specific foods, diets, or supplements that are effective in managing SLE. No single dietary approach has been proven to consistently influence disease activity. However, positive feedback from patients and caregivers highlights the need for further investigation into the impact of nutrition on SLE-related symptoms. Future interventional studies that account for socioeconomic and clinical characteristics are essential to identify dietary strategies that could improve SLE management.”
- The clarity of figure 1 is very much poor, increase its resolution
Thank you for your suggestion. We have provided a new figure (Figure 2) with higher resolution.